# Effect of Confinement on Anxiety Symptoms and Sleep Quality during the COVID-19 Pandemic

**DOI:** 10.3390/bs12100398

**Published:** 2022-10-17

**Authors:** Caren Alvarado-Aravena, Karem Arriaza, Matías Castillo-Aguilar, Karen Flores, Alexies Dagnino-Subiabre, Claudia Estrada-Goic, Cristian Núñez-Espinosa

**Affiliations:** 1School of Medicine, University of Magallanes, Punta Arenas 6210427, Chile; 2Institute of Health Studies, University Arturo Prat, Iquique 1100000, Chile; 3Institute DECIPHER, German-Chilean Institute for Research on Pulmonary Hypoxia and Its Health Sequelae, Iquique 1100000, Chile; 4Centro Asistencial de Docencia e Investigación (CADI-UMAG), University of Magallanes, Punta Arenas 6210427, Chile; 5Kinesiology Department, University of Magallanes, Punta Arenas 6210427, Chile; 6Laboratory of Stress Neurobiology, Centro de Neurobiología y Fisiopatología Integrativa (CENFI), Institute of Physiology, Faculty of Sciences, Universidad de Valparaiso, Valparaíso 2360102, Chile; 7Psychology Department, University of Magallanes, Punta Arenas 6210427, Chile

**Keywords:** anxiety, sleep quality, COVID-19, mental health, confinement

## Abstract

Confinement during the COVID-19 pandemic has significantly impacted lifestyles worldwide. The aim of this study was to evaluate the effect of confinement on anxiety symptoms and sleep quality in people living in extreme southern latitudes. The Beck Anxiety Inventory (BAI) and the Pittsburgh Sleep Quality Index (PSQI) were administered to 617 people, 74.2% of whom were women. The sample was grouped according to confinement: the zone of confinement (CZ) (46.5%) and the zone of partial confinement (PZ) (53.5%). In addition, the sample was further categorized into four age subgroups (18–25 years; 26–40 years; 41–50 years; over 50 years). Higher levels of anxiety and worse sleep quality were found in the CZ group than in the PZ group. Women had higher levels of anxiety and worse sleep quality than men. A significant bidirectional relationship between anxiety and sleep quality was observed, even after controlling for sex. This study demonstrated that women and young adults were more vulnerable to the effects of confinement on anxiety symptoms and sleep quality in populations at southern latitudes.

## 1. Introduction

The global outbreak of severe acute respiratory syndrome coronavirus 2 (SARS-CoV-2) has led to changes in lifestyle due to health restrictions and periods of confinement; these changes have affected the health and quality of life of people worldwide [1]. The rapid spread of coronavirus disease 2019 (COVID-19) has led governments to implement different strategies to prevent high mortality; as a result, populations have experienced periods of confinement and isolation [2]. During periods of social isolation, mental health was substantially impacted in addition to economic and health factors [3,4,5]. Prior to the pandemic, in the region of the Americas, 21% of the population presented some anxiety disorder [6]; however, there is no information on the prevalence of anxiety disorders in the Chilean population. The negative psychological impact of the pandemic worldwide [7] increased anxiety [8,9,10]. In addition to the neurobiological impact of these changes on the central nervous system [11], the pandemic itself increased people’s vigilance and perceived stress, leading to sleep disturbances [12,13,14]. Moreover, both anxiety and sleep quality play a key role in immune function [15,16,17]. The influence of these factors on the immune system is directly related to the response to respiratory infections; thus, the infection rate is lower in those who have better sleep quality and lower rates of anxiety [17,18,19,20]. Findings from a recent systematic review highlighted a high prevalence of sleep disturbances in the general population, accounting for up to 81% of individuals [21]; however, there is also no specific information on the Chilean population.

Confinement periods that limit interpersonal interactions, which alleviate stress and uncertainty, increased the frequency or severity of anxiety symptoms and decreased sleep quality [22]. In addition, age and sex influence these symptoms; specifically, these symptoms are worse in those under 35 years of age and in females [23].

In the high-latitude population, these relationships are also affected by social distancing caused by geographical conditions. This social distance directly affects the biopsychosocial functioning of individuals [24,25,26]. Additionally, younger individuals who live at high latitudes are affected by extreme changes in isolation between seasons, facilitating the development of psychological conditions that directly impact mental health [27,28,29]. In the southern latitudes, the increase in anxiety symptoms in young subjects is correlated with the increase in depressive symptoms; both symptoms are aggravated by pandemic-induced confinement and affect the mental health of subjects living in these geographical areas [30].

To date, no studies have evaluated these parameters in southern latitude populations during the pandemic. This study hypothesized that increased exposure to confinement would negatively affect anxiety symptoms and sleep quality among people who live in southern latitudes. Therefore, the main objective of this study was to determine the effect of confinement on anxiety and sleep quality in people who live in southern latitudes during the COVID-19 pandemic; our second aim was to determine the status of these same variables in the study population.

## 2. Materials and Methods

### 2.1. Design and Participants

This was a descriptive, observational, cross-sectional study. Using convenience sampling, we surveyed a representative number of participants within the same geographical area in southern Chile. A total of 617 individuals (age: median = 34 years, IQR = 22 years) participated in this study, of whom 458 (74.2%) were females. Of these individuals, 287 (46.5%) lived in a confined zone (CZ) all week; this region included Magallanes and the Chilean Antarctic. The remaining 330 subjects (53.5%), lived in partial confinement (PZ), with confinement lifted between Monday and Friday and imposed only on weekends; this region included Aysén. Both regions of Chile have similar demographic and population characteristics. In Chile, curfews and special permits granted by the police were used from the beginning of the pandemic until 1.5 years after its initial outbreak. Permits could be requested through the police station’s website, where one could apply for the individual temporary permit (ITP) for a specific reason (e.g., purchase of basic supplies, hospital attendance, departure of persons with an autism spectrum disorder, or other cognitive disabilities, or funeral attendance) [31]. In accordance with the confinement zones, two ITPs were available in CZs but could not be used after curfew; one ITP for weekend use was available in PZs but likewise could not be used after curfew.

In addition, participants were categorized according to age into four groups: participants 18–25 years old (28.8%), 26–40 years old (34.4%), 41–50 years old (20.9%), and over 50 years old (15.9%). The inclusion criteria for participants were as follows: 18 years of age or older and Chilean nationality. The exclusion criteria were as follows: incomplete questionnaires, use of sleeping medication, and moving from a confined area to a nonconfined area during the assessments. The study was approved by the Research Ethics Committee of the Universidad de Magallanes (no. 025/CEC/2020) and conducted in accordance with the Declaration of Helsinki.

### 2.2. Assessments

#### 2.2.1. Anxiety

Anxiety was determined using a Spanish version of the Beck Anxiety Inventory (BAI) [32]. This instrument collects self-report data; it has 21 items with scores on a Likert-type scale ranging from 0 to 3. BAI scores can be used to classify anxiety as follows: 0–7 indicates no anxiety, 8–15 indicates mild anxiety, 16–25 indicates moderate anxiety, and 26–63 indicates severe anxiety [33].

#### 2.2.2. Sleep Quality

The validated Spanish version of the Pittsburgh Sleep Quality Index (PSQI) was used [34]. This instrument collects self-report data on the following subscales: subjective sleep quality, sleep latency, sleep duration, habitual sleep efficiency, sleep disturbances, use of sleep medication, and daytime dysfunction. The score on each subscale ranges from 0 to 3, with 0 = very good; 1 = good; 2 = poor; and 3 = very poor. The total score is the sum of the seven subscales and ranges from 0 to 21 points. Total scores less than or equal to 5 indicate good sleep quality, while scores above 5 are interpreted as poor sleep quality.

### 2.3. Procedure

Data were obtained from an online questionnaire administered from 14 September to 12 October 2020 during the COVID-19 pandemic, which was delivered through different social networks generating a snowball sampling. The same form was used to collect simultaneous responses from both types of zones (CZ and PZ). The data were collected by the researchers in electronic databases for later analysis.

After giving informed consent, participants provided a range of sociodemographic data, including their age, sex, and current region of residence (to determine whether quarantine measures were in place). Other self-report data included the presence of clinically diagnosed anxiety and/or a sleep disorder and the use of pharmacological treatment (anxiolytics and/or sleep medication). Finally, anxiety and sleep quality surveys were administered with multiple-choice or drop-down responses, depending on the platform. Participants were informed whom to consult if they had any questions. Those who did not answer the survey in its entirety or who lived in other regions of Chile were not considered part of the sample.

### 2.4. Statistical Analysis

Continuous variables are expressed as the median and the interquartile range (*IQR*), and categorical variables are expressed as the absolute (n) and relative (%) frequencies. Groups were compared using nonparametric statistics due to the nonnormal distribution of the variables, which was verified by applying the *Kolmogorov–Smirnov* test.

To evaluate group differences in PSQI and BAI scores, the *Wilcoxon* rank sum test ($\ln\left(W\right)$) was used, while the *Kruskal–Wallis* rank sum test ($\chi_{|Kruskal-Wallis}^{2}$) was used to compare the mean ranges between age groups. To determine the independence of factor pairs, *Pearson’s* chi-square test ($\chi_{|Pearson}^{2}$) was used.

To explore the effects of confinement on anxiety and sleep quality, we constructed a linear model (analysis of variance; ANOVA) controlling for the influence of sex; a square root transformation was applied to the dependent variables (PSQI and BAI scores; $\sqrt{PSQI}$ and $\sqrt{BAI}$) to achieve a normal distribution of the residuals and homogeneity of variance, assessed with *Levene’s* test.

To determine the effect size (ES), a biserial correlation was used for the *Wilcoxon* test (${\hat{r}}_{biserial}$), an epsilon square was used for the *Kruskal–Wallis* test (${\hat{\epsilon }}^{2}$) and the partial eta-square ($\eta_{p}^{2}$) was used for the ANOVA.

The type I error level (α) was set at less than or equal to 5% (i.e., *p* ≤ 0.05 indicated statistical significance). The statistical analysis was carried out using the R programming language (version 4.1.1) in the RStudio graphical interface [35,36].

To evaluate the statistical power (1-β) of our study, we used G*Power. With our sample size (n = 617) and significance level (α = 0.05), we observed that the power to detect small (d = 0. 3) and moderate (d = 0.5) ESs was 95.2% and 99.9%, respectively, for comparisons of the confinement zone. For comparisons of sex differences, the power to detect the same ESs was 88.9% and 99.9%, respectively. For the ANOVA, the power to detect a small ES was 93.5% (f = 0.14, equivalent to an $\eta_{p}^{2}$ of 0.02).

To evaluate the internal consistency of the PSQI and BAI in our data, we used the standardized Cronbach’s alpha (α), calculating its 95% confidence interval (CI) via nonparametric bootstrap resampling with 10,000 replicates using the R package *ltm* [37]. The BAI demonstrated excellent internal consistency (α = 0.931, 95% CI [0.92, 0.94]), and the PSQI demonstrated good internal consistency (α = 0.801, 95% CI [0.78, 0.82]).

## 3. Results

Regarding anxiety symptoms, we found that the CZ group (median = 9, *IQR* = 13) exhibited significantly greater anxiety than the PZ group (median = 5, *IQR* = 9), $\ln\left(W\right)$ = 11.01, *p* < 0.001, ${\hat{r}}_{biserial}$ = 0.28, 95% CI [0.19, 0.36]. The same pattern was observed regarding sleep quality, with the CZ group reporting significantly higher mean PSQI scores (median= 9, *IQR* = 6) than the PZ group (median= 8, *IQR* = 6), $\ln\left(W\right)$ = 10.89, *p* = 0.006, ${\hat{r}}_{biserial}$ = 0.13, 95% CI [0.04, 0.22].

Regarding the effect of sex on anxiety symptoms, we observed that females scored higher than males on the BAI (females: median = 7, *IQR* = 12; men: median = 5, *IQR* = 9), $\ln\left(W\right)$ = 10.69, *p* < 0.001, ${\hat{r}}_{biserial}$ = 0.21, 95% CI [0.11, 0.31]. Sex was also associated with quality of sleep (PSQI scores), $\ln\left(W\right)$ = 10.7, *p* < 0.001, ${\hat{r}}_{biserial}$ = 0.22, 95% CI [0.12, 0.32]. However, after adjusting for the confinement area, we found that the effect of sex on anxiety was greater in the CZ group (${\hat{r}}_{biserial}$ = 0.23, 95% CI [0.08, 0.38]) than the PZ group (${\hat{r}}_{biserial}$ = 0.15, 95% CI [0.02, 0.28]). The same pattern was observed in terms of sleep quality, with females exhibiting higher PSQI scores than males (females: median = 9, *IQR* = 6; men: median = 7, *IQR* = 5.5), $\ln\left(W\right)$ = 10.70, *p* < 0.001, ${\hat{r}}_{biserial}$ = 0.22, 95% CI [0.12, 0.32]. When we controlled for confinement, we observed differences in the magnitude of the effect of sex on sleep quality; the effect of sex on sleep quality was greater in the PZ group (PZ: ${\hat{r}}_{biserial}$ = 0.22, 95% CI [0.09, 0.35]; CZ: ${\hat{r}}_{biserial}$ = 0.19, 95% CI [0.03, 0.34]). The rest of the sociodemographic characteristics can be seen in Table 1.

The impact of confinement on the transformed dependent variables (BAI and PSQI scores) is shown in Table 2. The bivariate distribution and the quantitative nature of the transformed variables according to confinement zone (CZ and PZ) are displayed in Figure 1.

To analyze differences in the effect of variables on $\sqrt{PSQI}$ and $\sqrt{BAI}$, we implemented six different multiple linear regression models. The simple model for the effect of confinement zone on $\sqrt{PSQI}$ with sex as a covariate (Model 1_P_) achieved better fit (*F*(1, 615) = 4.51, *p* = 0.034) than the simple model without sex as a covariate (Model 0_P_); the same pattern was observed for simple models for the effect of confinement zone on $\sqrt{PSQI}$ (Model 1_B_ vs. Model 0_B_) (*F*(1, 615) = 5.84, *p* = 0.016). The full (final) models including both sex and age group as covariates (Model 2_P_ and Model 2_B_) outperformed Model 0_B_ and Model 1_B_ (*F*(4, 615) = 21.51, *p* < 0.001 and *F*(3, 614) = 26.98, *p* < 0.001, respectively), as well as Model 0_P_ and Model 1_P_ (*F*(4, 615) = 4.72, *p* < 0.001 and *F*(3, 614) = 4.32, *p* = 0.005, respectively). The final effects for Model 2_P_ and Model 2_B_ are shown in Figure 2.

## 4. Discussion

At present, more than two years after the start of the COVID-19 pandemic, researchers have aimed to assess the impact of confinement on mental health [22,38]. The vast majority of studies that have investigated the effect of social isolation on mental health in the general population analyzed pre- and post-infection outcomes or compared their results with pre-COVID-19 data [23,39,40]. In this study, simultaneous comparisons were performed in southern latitudes. Our results indicate that stricter confinement was significantly associated with increased anxiety and worse quality of sleep in the general population. Although these results are similar to those of studies conducted in European and Asian countries [41,42], data confirming these effects in populations living in southern latitudes are lacking.

Other studies have shown that confinement is strongly related to social distancing, a strong human stressor [43,44], possibly explaining some of the observed sex differences. In line with this idea, females reported a higher rate of anxiety than their opposite-sex peers in the CZ group but not in the PZ group, consistent with findings in Austria and Brazil [23,45]. This sex difference might stem from the higher risk for various mental health conditions in females due to heredity, hormonal effects, and environmental factors [46,47].

The age of the participants was also an important factor. The prefrontal cortex (PFC) modulates the neuronal activity of the amygdala, a key brain area for the regulation of anxiety and fear [48]. The PFC is highly involved in dendritic development and synaptogenesis, which have different effects on psychological responses depending on the subject’s age [49]. The youngest age group in this study (18–25 years) presented higher anxiety levels than the other groups. Therefore, it is possible that PFC control over the amygdala is still developing in these young adults.

The brains of young adults may be more vulnerable to anxiety. Another possible explanation may be the lifestyle of young adults. The majority of this age group is likely attending university, where the pandemic has induced additional stress due to rapid changes (e.g., online classes and the uncertainty of academic semesters) [50]. In addition, the use of smart devices in young adults (between the ages of 18 and 25 years) is associated with higher levels of loneliness, which, in turn, are associated with higher levels of anxiety [50]. In the present study, confinement led to anxiety, even after controlling for sex and age. This finding further confirms the key role of confinement in the manifestation of anxiety symptoms and impaired mental health of people living in southern latitudes.

Regarding the quality of sleep, the sex differences were similar to those found in terms of anxiety. Females had the highest PSQI scores, which translates into worse sleep quality; after controlling for confinement zones, sleep quality was worse in females in the PZ group but not the CZ group. We speculate that although there was a change in the role of women in society, their work and role within the home led to emotional overload and poor sleep quality [22,51]. Stricter confinement might alleviate some of the demands of these roles, partially alleviating their effects on women. These results are quite similar to other studies, independent of the period of the COVID-19 pandemic, which also found a decrease in sleep quality in the population and, specifically, in women [52,53]. Therefore, these findings confirm that confinement reduces sleep quality, even after controlling for sex, in the southern latitude population studied. However, when the statistical analysis also controlled for participant age, the effect of confinement on sleep quality was no longer significant. These findings show that, unlike anxiety, sleep quality is more dependent on the age of subjects. Therefore, the effect of confinement may differ among age groups, resulting in a lower impact of confinement on the sleep quality of younger people.

Some factors, such as the composition of the nuclear family, the location and type of housing, the use of alcohol and other substances, and the presence of different social stressors, such as labor and domestic violence, were not controlled in this study. These confounds might directly affect anxiety symptoms and quality of sleep. Additionally, the online nature of the questionnaire might have induced erroneous interpretations of some survey items, since questions could not be resolved immediately during the completion of the questionnaires. On the other hand, it was not determined whether any part of the study population was infected by COVID-19, which may be a factor influencing the results on the basis of the infected versus uninfected population. Finally, future research on this topic should focus on longitudinal and experimental designs, ideally ensuring a homogeneous population with random sampling. In addition, since this study was performed in an area with extreme latitudes, the psychological impacts of isolation may vary according to exposure to extreme cold or heat.

## 5. Conclusions

Our results suggest that among individuals living at extreme southern latitudes, three main factors increase anxiety and reduce sleep quality: (1) confinement status: people who lived in confined zones were more affected; (2) sex: women were more vulnerable than men; and (3) age: young adults were more affected than the other age groups. Taken together, our results support the hypothesis that confinement negatively impacts anxiety symptoms and sleep quality in people living at southern latitudes. The results of this study can inform strategies to help people cope with the effects of confinement on anxiety and sleep quality in southern latitudes.

## Figures and Tables

**Figure 1 behavsci-12-00398-f001:**
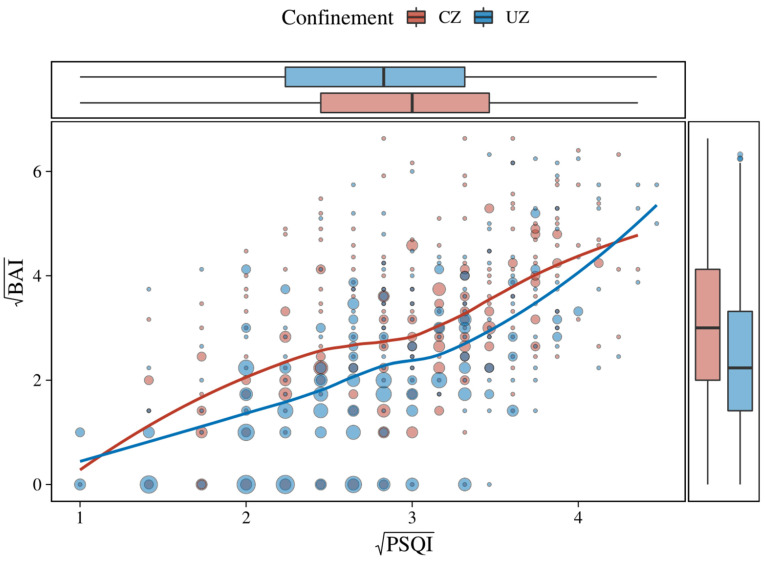
Scatter plot of $\sqrt{PSQI}$ and $\sqrt{BAI}$ with superimposed weighted local regression lines (LOESS); box plots indicate the distributions of these scores for the CZ and PZ groups. The size of the points represents the number of observations at each point.

**Figure 2 behavsci-12-00398-f002:**
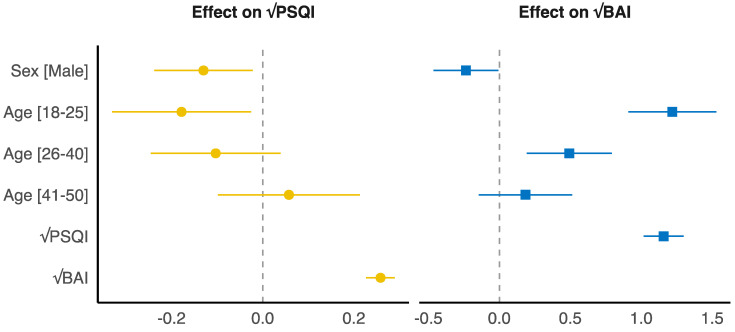
Model 2_P_ (**left**) and Model 2_B_ (**right**) explain significant and substantial proportions of the variance (R^2^ = 0.39, *F*(5, 611) = 78.92, *p* < 0.001, adjusted R^2^ = 0.39).

**Table 1 behavsci-12-00398-t001:** Sociodemographic characteristics according to area, age group, and psychometric parameters.

	Variable	Age Groups				
		18–25 Years n = 178 (28.8%)	26–40 Years n = 212 (34.4%)	41–50 Years n = 129 (20.9%)	>50 Years n = 98 (15.9%)	*p Value*
CZ	Sex					0.814
	Female	113 (18%)	63 (10%)	28 (4%)	21 (3%)	
	Male	28 (4%)	17 (3%)	10 (2%)	7 (1%)	
	BAI score	13 (15) ^x^	9 (11.3) ^z^	7 (7)	3.5 (6.3)	<0.001
	PSQI score	10 (5)	8.5 (6)	10 (6)	8 (6.3)	0.075
PZ	Sex					0.808
	Female	25 (4%)	94 (15%)	71 (11%)	49 (8%)	
	Male	13 (2%)	40 (6%)	25 (4%)	21 (3%)	
	BAI score	9 (12) ^x^	4 (11)	4.5 (8)	4 (7)	0.017
	PSQI score	7 (4)	8 (6)	8.5 (6)	8 (5)	0.519

Data are presented for each age range as the median (*IQR*) for BAI and PSQI scores. For the variables sex and group (CZ, confinement; PZ, partial confinement), data are expressed as n (%). The p value of each variable corresponds to *Pearson’s* χ^2^ test or a *Wilcoxon* rank sum test; x indicates *p* < 0.05 compared to the other groups; z indicates *p* < 0.05 compared to the >50-year age group.

**Table 2 behavsci-12-00398-t002:** ANOVA results for the influence of sex, zone, and age on the BAI and PSQI scores (transformed).

		$\sqrt{BAI}$							$\sqrt{PSQI}$						
Model	Parameter	SS	*df*	MS	*F*	*p*	$\eta_{p}^{2}$	95% CI	SS	*df*	MS	*F*	*p*	$\eta_{p}^{2}$	95% CI
1	Zone	80.74	1	80.74	33.4	<0.001	0.05	[0.03, 1.00]	3.72	1	3.72	7.34	0.007	0.01	[0.00, 1.00]
	Residuals	1486.69	615	2.42					312.24	615	0.51				
2 ^a,b^	Sex	42.54	1	42.54	17.97	<0.001	0.03	[0.01, 1.00]	9.24	1	9.24	18.66	<0.001	0.03	[0.01, 1.00]
	Zone	71.22	1	71.22	30.08	<0.001	0.05	[0.02, 1.00]	2.78	1	2.78	5.61	0.018	0.01	[0.00, 1.00]
	Sex × Zone	0.51	1	0.51	0.22	0.643	0.00	[0.00, 1.00]	0.04	1	0.04	0.08	0.774	0.00	[0.00, 1.00]
	Residuals	1453.16	613	2.37					303.91	613	0.50				
3 ^a^	Sex	42.54	1	42.54	19.31	<0.001	0.03	[0.01, 1.00]	9.24	1	9.24	18.72	<0.001	0.03	[0.01, 1.00]
	Age _cat_	166.87	3	55.62	25.25	<0.001	0.11	[0.07, 1.00]	3.80	3	1.27	2.57	0.054	0.01	[0.00, 1.00]
	Zone	11.83	1	11.83	5.37	0.021	0.01	[0.00, 1.00]	1.40	1	1.40	2.86	0.092	0.00	[0.00, 1.00]
	Sex × Age _cat_	2.37	3	0.79	0.36	0.782	0.00	[0.00, 1.00]	1.35	3	0.45	0.92	0.432	0.00	[0.00, 1.00]
	Sex × Zone	0.33	1	0.33	0.15	0.701	0.00	[0.00, 1.00]	0.02	1	0.02	0.04	0.833	0.00	[0.00, 1.00]
	Age _cat_ × Zone	14.91	3	4.97	2.26	0.080	0.01	[0.00, 1.00]	3.03	3	1.01	2.05	0.105	0.01	[0.00, 1.00]
	Residuals	1328.59	604	2.20					297.12	604	0.49				
Compared Levels		Difference		95% CI		SE	*t*(613)	p ^†^	Difference		95% CI		SE	*t*(613)	*p* ^†^
Female CZ	Female PZ	0.72		[0.34, 1.10]		0.14	4.98	<0.001	0.13		[−0.05, 0.30]		0.07	1.91	0.225
Female CZ	Male CZ	0.61		[0.02, 1.19]		0.22	2.76	0.030	0.24		[−0.02, 0.51]		0.10	2.42	0.075
Female CZ	Male PZ	1.19		[0.70, 1.69]		0.19	6.38	<0.001	0.41		[0.18, 0.63]		0.09	4.76	<0.001
Female PZ	Male PZ	0.48		[−0.02, 0.97]		0.19	2.56	0.053	0.28		[0.06, 0.51]		0.09	3.31	0.005
Male CZ	Female PZ	0.11		[−0.47, 0.69]		0.22	0.49	0.961	−0.12		[−0.38, 0.15]		0.10	−1.18	0.642
Male CZ	Male PZ	0.58		[−0.08, 1.25]		0.25	2.33	0.093	0.16		[−0.14, 0.47]		0.11	1.43	0.482

Pairwise comparisons are made to assess the differences between subgroups within confinement groups in order to evidence the role of sex adjusting by the influence of different confinement levels in the transformed variables of anxiety and sleep quality. The parameters are zone (confinement [CZ] or partial confinement [PZ]), sex (female or male) and age _cat_ (age group: 18–25 years; 26–40 years; 41–50 years; or >50 years). ^a^, significant differences (*p* < 0.001) of BAI scores compared to the previous model; ^b^, significant differences (*p* < 0.001) of PSQI scores compared to the previous model; ^†^, false discovery rate (FDR)correction on p values in the post hoc comparisons.

## Data Availability

The data generated and used in this study are available in the form of an R package called *AnxietySleep.* Information regarding its installation and use can be found at https://nim-ach.github.io/AnxietySleep. Last update on 3 June 2022.

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
