# Peer review of "Effect of Confinement on Anxiety Symptoms and Sleep Quality during the COVID-19 Pandemic"

_behavsci, 2022, doi:10.3390/bs12100398_

Round 1
Reviewer 1 Report
The manuscript shows a survey related to anxiety and quality of life during Covid-19 pandemic. The strengths of the study are the sample size and the weakness are the procedure explanation and the novelty of the topic.
I have some considerations:
Introduction: some information about general characteristics of anxiety and quality of life from Chile should be added.
Line 89: Sample size: there are two groups CZ and PZ and 4 subgroups. These four subgroups included both mentioned groups together (CZ and PZ) or are analyzed separately.
Are there any criteria for the age ranges?
Have concomitant diseases that could affect mental health been considered?
Line 112. Procedure: How the sample was contacted? Who collect this data? All people contacted respond the survey? Is there any drop out? How many researchers are involved in this study?
Line 184: table 2: table are huge and it is a bit confusing, considering the amount of information.
Discussion: some information about quality of life should be added, although the result of this study are not conclusive, they should be compared with other published articles.
Author Response
Response to Reviewer 1 Comments
Point 1: Introduction: some information about general characteristics of anxiety and quality of life from Chile should be added.
Response 1: We appreciate the comment. In fact, this type of information did not exist. Although no studies relate the same variables as this study, we have included information on what was requested in lines 41, 42 and 51-53, which enriched the work..
Point 2: Line 89: Sample size: there are two groups CZ and PZ and 4 subgroups. These four subgroups included both mentioned groups together (CZ and PZ) or are analyzed separately.
Response 2: The analyzes are carried out considering the structure of the sampling, executing the analyzes on age groups separated by state of confinement (CZ and PZ).
Point 3: Are there any criteria for the age ranges?
Response 3: The cutoff width was set to allow equal probability density for each group relative to the whole sample size, and in this way, increase the statistical power in our analyses.
Point 4: Line 112. Procedure: How the sample was contacted? Who collect this data? All people contacted respond the survey? Is there any drop out? How many researchers are involved in this study?
Response 4: We appreciate the comment. We have supplemented the information on lines 114-115, 118-119 and 123-128. On the other hand, the work carried out by each researcher is described after the conclusion, in the section “Authors' contribution”.
Point 5: Line 184: table 2: table are huge and it is a bit confusing, considering the amount of information.
Response 5: Table 2 is necessary to support our claims in the discussion section. However, an additional informative footer has been added to the bottom of the table, indicating the primary purpose of the pairwise comparisons section. If the reviewer still feels that the table is too large for publication, we as authors, are more than willing to implement additional modifications or deletions to the table.
Point 6: Discussion: some information about quality of life should be added, although the result of this study are not conclusive, they should be compared with other published articles.
Response 6: We appreciate this contribution to our work. We have added other references that compare our results with what exists in the current literature that enriches the discussion. They are inserted on lines 251 to 253.

Reviewer 2 Report
The manuscript reports the effect of confinement on anxiety and sleep during the COVID-19 pandemic. This is certainly a timely and interesting work. Overall, the manuscript has clear logic appropriate data interpretation and is well written. This work brings insights from the real world onto questions that have been mostly concerned. Therefore, I believe that this work is highly suitable for publication in the present situation. Some comments are given below:
1. Due to the widespread and multisystem side effects of COVID-19, the health condition of participants should be considered in this study. Whether all individuals never got COVID-19 or recovered? More information needs to be added to make the conclusion more convincing.
2. Fig1, what are two blue points mean in the box plot?
3. Line161-164, redundant description.
Author Response
Response to Reviewer 2 Comments
Point 1: Due to the widespread and multisystem side effects of COVID-19, the health condition of participants should be considered in this study. Whether all individuals never got COVID-19 or recovered? More information needs to be added to make the conclusion more convincing.
Response 1: This point is very interesting. However, we did not ask for this information in our questionnaire, so we added it as a limitation of our study on lines 262-264.
Point 2: Fig1, what are two blue points mean in the box plot?
Response 2: Represent the non-parametric outliers, (i.e., those with absolute values greater than 3 times the interquartile range)
Point 3: A Line161-164, redundant description.
Response 3: We agree with this statement. Line 163-164 was removed

Round 2
Reviewer 1 Report
Dear authors,
I agree with the modifications you have made. Table 2: better with the footnote. Thank you for considering my suggestions.
Good job.